# MULTI-MODALITY ADVERSARIAL ATTACKS ON LATENT DIFFUSION MODELS

## ABSTRACT

Latent diffusion models (LDMs) achieve unprecedented success in image editing, which can accurately edit the target image with text guidance. However, the multi-modal adversarial robustness of latent diffusion models has not been studied. Previous works only focus on single modality perturbation, such as image or text, making them less effective while more noticeable to humans. Therefore, in this paper, we aim to analyze the multi-modal robustness of latent diffusion models through adversarial attacks. We propose the first Multi-Modality adversarial Attack algorithm (MMA), which modifies the image and text simultaneously in a unified framework to determine updating text or image in each step. In each iteration, MMA constructs the perturbed candidates for both text and image based on the input attribution. Then, MMA selects the perturbed candidate with the largest $L_2$ distortion on the cross attention module in one step. The unified query ranking framework properly combines the updating from both modalities. Extensive experiments on both white-box and black-box settings validate two advantages of MMA: (1) MMA can easily trigger the failure of LDMs (high effectiveness). (2) MMA requires less perturbation budget compared with single modality attacks (high invisibility).

## 1 INTRODUCTION

Latent diffusion models (LDMs) (Rombach et al., 2022) are widely deployed in a variety of image synthesis and editing applications, such as DALL·E2(Ramesh et al., 2022) and Stable Diffusion (Rombach et al., 2022) due to their efficiency and exceptional generation quality. Latent diffusion models generate real images from the latent noise sampled from a uniform Gaussian distribution through an iterative denoising process in the latent space. During the denoising process, additional information, such as text or image, can be fused on the sampled latent noise to guide the image generation through the conditioning mechanism. Specially, to fulfill the function of image editing, latent diffusion models combine the embedding of the text prompt and a given image through cross attention.

However, recent research (Liu et al., 2023; Salman et al., 2023) reveals latent diffusion models are vulnerable to adversarial attacks, which modify the input in a human-imperceptible way so that latent diffusion models will generate low-quality images or incorrectly generate the input image. With the aim of analyzing the robustness issues and training better diffusion models, it is of great significance to design effective adversarial attack algorithms.

We categorize recent adversarial attacks on diffusion models into two branches based on their modality: textual adversarial attacks (Zhuang et al., 2023) and image adversarial attacks (Salman et al., 2023). Recent textual adversarial attacks add meaningless suffixes to the input text by attacking the text encoder (Zhuang et al., 2023) or utilizing the gradient guidance from an image classifier (Liu et al., 2023). Image adversarial attacks perturb the image encoder (Salman et al., 2023) or the intermediate modules (Zhang et al., 2023b) for generating the imperceptible noise on the input image. All of the previous approaches only modify one modality of the inputs (text or image), which cannot fully explore the input space for generating effective perturbation. Additionally, the perturbation magnitude of single-modality attacks is high, making them noticeable to humans. For example, the added meaningless tokens of textual adversarial attacks are not daily words, and the number of added tokens is large (Zhuang et al., 2023). Therefore, for the sake of devising effective and imperceptible

adversarial attacks, we propose to explore multi-modality adversarial attacks where we modify the input text and image jointly. Perturbing the text and image at the same time distributes the needed perturbation to both modalities, making them imperceptible to humans on the whole. Besides, compared with single-modality attacks, multi-modality adversarial attacks have the potential to discover more robustness issues and thus contribute to a better understand of the working principles of LDMs.

In this paper, we propose the Multi-Modality Attack (MMA) to update the text and image in a unified framework for generating adversarial samples of latent diffusion models. MMA includes a unified query ranking framework to select to perturb text or image in each update step. First, MMA attributes the output of the LDM to both input text and image. Then, MMA constructs the perturbed candidates for both text and image based on the attribution results. Finally, MMA selects the perturbed candidate with the largest $L_2$ distortion on the cross attention module in one step. The unified query ranking framework properly combines the updating from both modalities. Unlike previous works, we constrain the perturbation space to enhance adversarial imperceptibility. Specifically, we only consider token replacement with its synonym for text perturbation and small $L_2$ norm image noise for image perturbation. Comprehensive experiments validate the effectiveness and imperceptibility of the proposed multi-modality adversarial attacks.

The contributions of our work are threefold:

- Current adversarial attacks on diffusion models are based on single-modality perturbation, which makes them less effective and more noticeable to humans. Therefore, we formally define the multi-modality adversarial attacks on the latent diffusion model and propose the first multi-modality adversarial attack on LDMs called MMA.

- MMA contains a query ranking framework to update input text or image iteratively. In each update iteration, MMA attributes the output to both input text and images to construct replacement candidates. MMA ranks the candidates by querying the distortion of the cross attention module in LDM for updating. Such a framework properly combines the perturbation of text and image.

- We conduct extensive experiments on both white-box and black-box settings to validate the effectiveness and imperceptibility of our multi-modality attack. Based on the experimental results and analysis, we conclude that combining text and image perturbations can more effectively mislead LDMs.

## 2 RELATED WORK

### 2.1 LATENT DIFFUSION MODELS

Diffusion models (Ho et al., 2020; Rombach et al., 2022) have achieved state-of-the-art generation performance recently, which can generate vivid images with the conditions of text and images. Diffusion models generate the image through a stochastic differential denoising process. The denoising process starts from an image sampled from a uniform Gaussian distribution and adds a small Gaussian noise on the image repeatedly to produce a real image. Latent diffusion models conduct the denoising process on the latent features instead of images to improve efficiency and reduce computation complexity. Furthermore, it is convenient to inject the conditions (text or image) into latent features to guide the generation process, making latent diffusion models come to the fore.

Image editing tasks require the diffusion models to edit the given image with the guidance of a text prompt. Based on the conditioning phenomenon, latent diffusion models initialize the latent feature by a combination of the given image feature and a random sampled Gaussian noise to implement image editing. During the denoising process, the latent features then interact with the text features via the cross attention module to edit the latent features. Finally, the latent features are decoded back to the image domain as the edited image. Especially, the encoder and decoder for the image domain is a variational autoencoder (Kingma & Welling, 2013), and the encoder for the text domain is the CLIP encoder (Radford et al., 2021). For simplicity, we call the pipeline of latent diffusion models for image editing tasks as img2img. Figure 1 illustrates the generation process of the img2img pipeline.

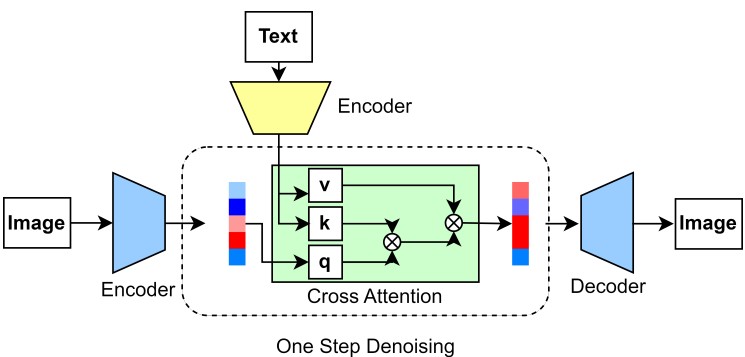

Figure 1: The workflow of img2img latent diffusion models for image editing, which combines the features of image and text through the cross attention module.

## 2.2 ADVERSARIAL ATTACKS

Deep neural networks, including vision models and language models, are known to be vulnerable to adversarial attacks (Dong et al., 2018; Li et al., 2020). Given the input $x$ and a DNN model $f(x)$, adversarial attacks discover the adversarial example $x^{adv}$ to mislead the DNN model, while the perturbation between the input and the adversarial example is human-imperceptible.

Image adversarial attacks perturb the pixel values under the constraint $\left\|x - x^{adv}\right\|_p < \epsilon$ to satisfy invisibility, where $\left\|\cdot\right\|_p$ represents the $L_p$ norm. In this paper, we utilize the $L_\infty$ norm here as a hard constraint to align with previous adversarial attack papers (Dong et al., 2018; Lin et al., 2019). Since the image space is continuous, multiple effective attack algorithms on image classification model (Goodfellow et al., 2014) have been proposed based on the gradient of the network for solving the following problem:

$$\max_{x^{adv}} \ J(x^{adv}, y) \quad s.t. \left\|x - x^{adv}\right\|_\infty < \epsilon, \tag{1}$$

where $y$ is the ground truth label for the image $x$, and $J(x, y)$ is the cross-entropy loss.

There are three levels of modification granularity of textual adversarial attacks: character-level, word-level, and sentence-level (Qiu et al., 2022). Character-level attacks generally insert, delete, flip, replace, or swap individual characters in the text. Word-level attacks generally add new words, remove words, or change words in the sentences. Sentence-level attacks usually insert new sentences or paraphrase the original sentences. Compared with sentence-level attacks, character-level and word-level attacks have higher attack success rates (Zeng et al., 2021). Besides, word-level attacks are more stealthy than character-level attacks, which often introduce typos (Ebrahimi et al., 2018). Therefore, in this work, we only consider the word-level perturbation.

Nevertheless, the model structure and parameters are hidden from users in the real world. Black-box attacks (Dong et al., 2018) attract the attention of researchers, where attackers cannot get access to the model information. Transfer-based attacks (Dong et al., 2018) are a kind of black-box attacks, which are famous for their efficiency. In transfer-based attacks, attackers craft adversarial samples on a surrogate model with a white-box attack algorithm and directly transfer the adversarial examples to the target models without the information of the target model. In addition, transfer-based attacks can be utilized to analyze the common defects among different models (Zhang et al., 2023a). Therefore, in this work, we also discuss the transferability of multi-modality attacks. Additionally, (Chen et al., 2023) utilizes diffusion models for adversarial attacks, and (Zhang et al., 2022) analyzes the robustness of VLM.

## 2.3 ADVERSARIAL ATTACKS ON DIFFUSION MODELS

Current adversarial attacks on diffusion models mainly focus on a single modality. (Zhuang et al., 2023) adds meaningless characters to the text to mislead the text encoder, and the distorted text

features influence the generation performance. (Liu et al., 2023) also combines meaningless suffixes to the text by the gradient guidance of a consecutive image classifier to the output of the diffusion model. (Salman et al., 2023) and (Zhang et al., 2023b) focus on perturbing the image by distorting the intermediate features of the latent diffusion models. However, previous works only focus on modifying single-modality, leaving the combinatorial searching space of images and text unexplored. More importantly, single-modality attacks always generate human perceptible adversarial samples. Textual adversarial examples have long and meaningless characters or suffixes, and image adversarial samples have noticeable artifacts. Therefore, in this paper, we propose the multi-modality adversarial attack to explore the combinatorial searching space and generate human invisible adversarial samples.

## 3 PROBLEM STATEMENT

In this section, we first clarify the problem statement of the multi-modality adversarial attacks on latent diffusion models. We formally define the multi-modality attacking problem by specifying the attacking objective, the perturbation space, and the perturbation budget.

We denote the latent diffusion models as $LDM(x, t)$ with the input image $x$ and text $t$. Then, the output of the latent diffusion model, $y = LDM(x, t)$, is the generated image to satisfy the requirement of image editing or generation. Previous image or textual adversarial attacks on classification tasks have explicit measurements, like attack success rates, to assess whether the adversarial attack is successful or not. For multi-modality adversarial attacks on diffusion models, we aim to find the adversarial input image $x^{adv}$ and text $t^{adv}$ to mislead the function of the latent diffusion model. $Oracle(y^{adv}, t) = False$, where $y^{adv} = LDM(x^{adv}, t^{adv})$ is the output image with adversarial input, and $Oracle(\cdot)$ is the human judgment oracle on the similarity between the generated image and input text.

After the determination of the adversarial objective, we formally discuss the perturbation space of the multi-modality adversarial attack. Following the previous works in the field of image adversarial attacks, we set a hard constraint on image perturbation by the $L_\infty$ norm. The $L_\infty$ norm constrains the maximum perturbation on the pixel values of the image. As a result, the image perturbation should satisfy $\left\| x - x^{adv} \right\|_\infty < \epsilon$, where $\epsilon$ is the perturbation budget of the image.

For the textual perturbation, we also settle a hard constraint to guarantee imperceptibility. Following the previous work (Huang et al., 2022), we consider the token-wise Levenshtein distance between the original text and the adversarial text. The Levenshtein distance (Levenshtein et al., 1966) measures the number of edits to achieve the adversarial text starting from the original one, including the replacement, addition, and deletion of the token. Thus, the textual perturbation should satisfy $lev(TN(t), TN(t^{adv})) < \gamma$, where $TN(\cdot)$ represents a text tokenizer, and $\gamma$ is the perturbation budget of text.

All in all, the goal of multi-modality attack is to mislead the diffusion models from a human perspective $Oracle(LDM(x^{adv}, t^{adv}), t) = False$, while we keep the adversarial image satisfying the $L_\infty$ constraint $\left\| x - x^{adv} \right\|_\infty < \epsilon$, and the adversarial text satisfying the token-wise Levenshtein distance constraint $lev(TN(t), TN(t^{adv})) < \gamma$.

## 4 MULTI-MODALITY ATTACK

Since $LDM(x, t)$ is a generative model, it is intractable to optimize the objective directly. Alternatively, we aim to deform the latent features inside latent diffusion models by adversarial input. Since the latent diffusion model decodes the latent for generating image, the generated image will be largely influenced, if we distort the latent features by the adversarial input. As we mentioned before, the cross attention module combines the features from the image and text together in the denoising process, which is the most vital part of Img2img diffusion models. Therefore, we assume that the function of the latent diffusion model will be misled, if we distort the latent features of the cross attention module.

Therefore, the goal of our multi-modality attack is to distort the latent features of the cross attention module for each step of the denoising process, while we keep the adversarial image satisfying the

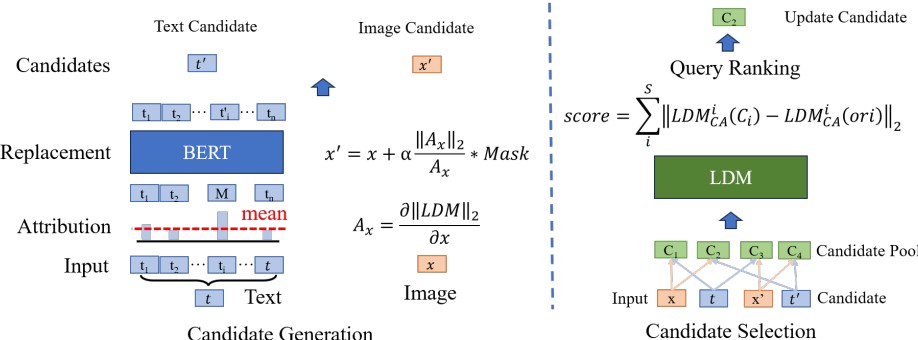

Figure 2: The framework overview of the proposed MMA. MMA employs a query ranking framework, consisting of two stages: candidate generation and candidate selection.

$L_\infty$ constraint and the adversarial text satisfying the token-wise Levenshtein distance constraint. The adversarial objective is formally defined as follows:

$$\max_{x^{adv}, t^{adv}} \sum_{i=1}^{S} \left\| LDM_{CA}^i(x^{adv}, t^{adv}) - LDM_{CA}^i(x, t) \right\|_2,$$
$$s.t. \; \left\| x - x^{adv} \right\|_\infty < \epsilon,$$
$$lev(TN(t), TN(t^{adv})) < \gamma, \tag{2}$$

where $S$ is the number of denoising steps, and $LDM_{CA}^i(\cdot)$ denotes the latent features of the cross attention module at the denoising step $i$.

We propose the Multi-Modality Attack (MMA) to achieve the above adversarial objective. MMA iteratively perturbs the input text and image to achieve the adversarial objective. The key part of MMA is to properly combine the text and image perturbations. For this purpose, MMA deploys a query ranking framework to update adversarial input. The overview of MMA is shown in Figure 2, containing two main components: candidate generation and candidate selection. MMA first generates perturbation candidates for both text and image in each iteration. Then, MMA selects the best perturbation candidate with feedback from the latent features to update the adversarial input. In MMA, the text perturbation and the image perturbation collaborate well for achieving the adversarial objective.

To be more specific, we suppose the input pair is $(x, t)$, and we update the input pair for $T$ iterations in total. In each iteration, we utilize the query ranking framework for generating the adversarial sample in one step, and feed in the generated example as the input for the next round. After running $T$ iterations, we finally obtain the adversarial example pair $(x^{adv}, t^{adv})$ by MMA.

## 4.1 CANDIDATE GENERATION

We first compute the contribution of the input image and text to the generated image via the attribution method (Sundararajan et al., 2017). Specifically, we employ the gradient of the $L_2$ norm of the generated image to the input as the attribution. We regard the input image as one token, so the attribution of the image token is:

$$A_x = \frac{\partial \left\| LDM(x, t) \right\|_2}{\partial x}, \tag{3}$$

We should be aware that the attribution has the same size as the input image, and the value measures the importance of the corresponding pixel on the generated image. Similarly, we also compute the attribution of the $i$-th text token by the following equation:

$$A_{t_i} = \left\| \frac{\partial \left\| LDM(x, t) \right\|_2}{\partial e(t_i)} \right\|_2, \tag{4}$$

where $e(\cdot)$ is the token embedding. Since the attribution on the text token is a vector, we use the magnitude of the vector as the numerical attribution of this token. Then, we can utilize the computed attribution as the guidance for generating replacement candidates.

### 4.1.1 TEXT PERTURBATION

We first explain the text perturbation in MMA. In order to keep the imperceptibility of the text perturbation, we only consider the token-level perturbation. Previous textual attacks add meaningless characters or suffixes to the input text for misleading diffusion models, which is human perceptive, and the semantic meaning of the text may be changed. Instead, we consider replacing the tokens in the text with their synonym to preserve semantic consistency and invisibility.

We get inspired by (Li et al., 2020), and we substitute the token with its synonym with the help of masked language models. The masked language models are trained with masked language loss, which has the ability to predict a masked token in a sequence, and the predicted token can fit the context well. Therefore, we utilize the masked language model, BERT (Kenton & Toutanova, 2019), to substitute the original tokens in the text.

Specifically, in each iteration, we mask out the tokens in the input text one by one with a masked token. Then, the masked language model BERT is utilized to predict the masked token. To guarantee the predicted token fits well in the text, we filter out low-confidence predictions. We also notice that some of the tokens are significant, and improper modification will change the total semantics of the text. Therefore, we utilize the previously computed token attribution to set an adaptive threshold for keeping the semantic consistency. We regard the token with high attribution as being important to the semantics, so we set a higher threshold of substitution for the tokens with higher attribution compared with the mean value. Similarly, we set a lower threshold of substitution for the remaining tokens. Therefore, the text candidate generation can be summarized in Algorithm 1 in the Appendix.

### 4.1.2 IMAGE PERTURBATION

As we treat the whole image as a special token, we aim to find the substitution of the image without sacrificing imperceptibility. We make use of the computed attribution on the image, which represents the contribution of the pixels to the output image. If we update the image along the direction of the attribution, the output image will be largely influenced. To guarantee invisibility, we utilize the $L_2$ attack for updating the image. Additionally, we introduce a random mask for image perturbation to balance the effectiveness of the attack and the imperceptibility of added noise. The random mask is sampled from a Bernoulli distribution with a probability of $p$. Therefore, the image candidates are represented in the following equation:

$$cand_{img}^s = x + \alpha \frac{A_x}{\|A_x\|_2} * Mask_s, \qquad (5)$$

where $\alpha$ is the step size for updating, and $s$ is the sampling time. The image candidate generation is shown in Algorithm 2 in Appendix.

### 4.2 CANDIDATE SELECTION

The next stage of MMA is to query the LDM and rank the generated candidates to pick the top one for updating. We concatenate the text candidates with the input image and image candidates with the input prompt as the candidate pool without crossly combining the candidates to reduce the perturbation. It means we only modify the text with one token or change the pixel value in one iteration. Thus, the candidate selection stage is to determine whether we update text or image in each iteration to combine the multi-modality perturbation properly.

As our adversarial objective is to distort the latent features of cross attention modules, we utilize the $L_2$ distortion of the latent features as the feedback score for each candidate. The score is computed by the following equation:

$$score(c) = \sum_{i=1}^{S} \left\| LDM_{CA}^i(x', t') - LDM_{CA}^i(x, t) \right\|_2, \qquad (6)$$

where $c = (x', t')$ is the candidate for the updating. We then rank the score and pick the best one as the final update in each iteration.

Table 1: The white-box performance of different attacking methods under various settings, including imperceptibility and effectiveness. The best results are in **bold**. ↓ indicates the lower the better, while ↑ represents the opposite.

| Model | Method | $L_2$↓ | $Sim_{img}$↑ | Lev↓ | $Sim_{text}$↑ | PSNR↓ | SSIM↓ | MS-SSIM↓ | CLIP↓ | FID↑ | IS↓ |
|---|---|---|---|---|---|---|---|---|---|---|---|
| SD1-5 | Image | 11.20 | 0.65 | - | - | 12.89 | 0.281 | 0.466 | 33.98 | 177.78 | **15.34** |
| | Text | - | - | 3.83 | 0.81 | 12.01 | 0.247 | 0.404 | **30.86** | 183.72 | 16.56 |
| | LDMR | 29.59 | 0.37 | - | - | 13.29 | 0.281 | 0.489 | 34.09 | 176.99 | 16.50 |
| | QF | - | - | 9.63 | **0.88** | 12.08 | 0.228 | 0.406 | 32.63 | 182.42 | 15.48 |
| | MMA | **8.43** | **0.71** | 3.46 | 0.84 | **11.17** | **0.185** | **0.337** | 31.16 | **185.96** | 16.47 |
| | MMA w/o Text | 8.43 | 0.71 | - | - | 13.44 | 0.321 | 0.508 | 34.29 | 178.13 | 16.30 |
| | MMA w/o Image | - | - | 3.46 | 0.84 | 12.22 | 0.256 | 0.416 | 31.30 | 182.03 | 16.57 |
| SD1-4 | Image | 11.24 | 0.65 | - | - | 12.90 | 0.289 | 0.481 | 33.93 | 177.87 | 15.92 |
| | Text | - | - | 3.84 | 0.81 | 12.00 | 0.245 | 0.405 | **30.59** | 187.63 | 16.55 |
| | LDMR | 29.51 | 0.38 | - | - | 13.22 | 0.279 | 0.489 | 34.11 | 176.80 | 15.72 |
| | QF | - | - | 9.63 | **0.88** | 12.20 | 0.234 | 0.411 | 33.35 | 184.79 | 16.06 |
| | MMA | **8.39** | **0.71** | 3.57 | 0.83 | **11.09** | **0.179** | **0.331** | 30.88 | **188.04** | 16.37 |
| | MMA w/o Text | 8.39 | 0.71 | - | - | 13.44 | 0.317 | 0.508 | 34.07 | 176.80 | **15.67** |
| | MMA w/o Image | - | - | 3.57 | 0.83 | 12.21 | 0.261 | 0.423 | 31.03 | 187.37 | 16.70 |
| SD2-1 | Image | 24.50 | 0.45 | - | - | 11.52 | 0.191 | 0.377 | 32.55 | 190.72 | 13.90 |
| | Text | - | - | 4.33 | 0.78 | 10.92 | 0.170 | 0.318 | 28.71 | 202.29 | **13.09** |
| | LDMR | 26.82 | 0.40 | - | - | 11.21 | 0.182 | 0.354 | 32.38 | 190.67 | 14.00 |
| | QF | - | - | 9.93 | **0.84** | 12.24 | 0.233 | 0.427 | 31.99 | 187.43 | 16.43 |
| | MMA | **16.51** | **0.56** | 3.78 | 0.80 | **10.14** | **0.123** | **0.263** | **28.65** | **205.06** | 13.38 |
| | MMA w/o Text | 16.51 | 0.56 | - | - | 11.80 | 0.213 | 0.400 | 32.49 | 189.85 | 13.63 |
| | MMA w/o Image | - | - | 3.78 | 0.80 | 10.99 | 0.184 | 0.333 | 28.97 | 201.41 | 13.46 |
| Pix2pix | Image | 12.00 | 0.64 | - | - | 12.34 | 0.170 | 0.392 | 33.83 | **166.50** | 15.39 |
| | Text | - | - | 3.70 | 0.83 | 9.25 | **0.008** | 0.030 | 33.21 | 123.97 | 16.21 |
| | LDMR | 28.07 | 0.385 | - | - | 13.16 | 0.166 | 0.416 | 33.57 | 126.74 | 15.26 |
| | QF | - | - | 9.63 | 0.88 | 15.28 | 0.490 | 0.657 | 34.12 | 121.61 | 15.40 |
| | MMA | **9.62** | **0.69** | 2.78 | **0.88** | **8.73** | **0.008** | **0.027** | **32.90** | 160.55 | **15.11** |
| | MMA w/o Text | 9.62 | 0.69 | - | - | 12.67 | 0.189 | 0.413 | 33.89 | 161.22 | 15.23 |
| | MMA w/o Image | - | - | 2.78 | **0.88** | 15.15 | 0.523 | 0.666 | 33.70 | 121.95 | 16.04 |

## 5 EXPERIMENT

### 5.1 EXPERIMENTAL SETUP

**Dataset:** We conduct experiments on the dataset proposed by (Zhang et al., 2023b), which includes 500 high-quality image and prompt pairs generated from COCO (Lin et al., 2014).

**Threat Model:** We consider four representative and widely used img2img latent diffusion models, including Stable Diffusion V1-4 (SD-v1-4) (Rombach et al., 2022), Stable Diffusion V1-5 (SD-v1-5), Stable Diffusion V2-1 (SD-v2-1), and Instruct-pix2pix (pix2pix) (Brooks et al., 2022) by following the experimental setting of (Zhang et al., 2023b). Stable diffusion models have a model structure different from the Instruct-pix2pix model. Different versions of diffusion models have the same structure, and the higher version is further trained on more training data.

**Baselines:** Since there are no works on the multi-modality adversarial attack on latent diffusion models, we first compare MMA with the single-modality adversarial attacks in our MMA framework. Specifically, we run MMA only on image or text modality that we drop off the text candidate generation module or image candidate generation module. We denote the baselines to be Image and Text, respectively. We also consider the ablations of MMA without one modality perturbation as the baselines. We suppose the input is $(x, t)$ and the adversarial sample generated by MMA is $(x^{adv}, t^{adv})$. We set MMA without image perturbation $(x, t^{adv})$ as MMA w/o Image, and MMA without text perturbation $(x^{adv}, t)$ as MMA w/o Text. Furthermore, we also compare MMA with two state-of-the-art single-modality adversarial attacks on diffusion models. LDMR (Zhang et al., 2023b) perturbs the image, while QF (Zhuang et al., 2023) adds meaningless suffixes to the prompts.

**Metric:** We evaluate the performance of adversarial attacks from two perspectives: imperceptibility and effectiveness. The imperceptibility measures the difference between the adversarial example and the benign input. The imperceptibility of image perturbation is represented by two metrics: (1) $L_2$ norm of the perturbation and (2) Structural Similarity Index Measure (SSIM) (Wang et al., 2004) score between the benign input and adversarial image. We also deploy two criteria to compute the imperceptibility of text perturbation: (1) token-wise Levenshtein distance (Levenshtein et al., 1966) and (2) semantic similarity between the embedding of the original and adversarial text encoded by

SentenceBERT (Reimers & Gurevych, 2020). We evaluate the effectiveness of the attack through six metrics: (1) Peak-Signal-to-Noise Ratio (PSNR) measures the ratio between the original generated image and the difference between the original generated image and the adversarial generated image. (2) Structural Similarity Index Measure (SSIM) (Wang et al., 2004) quantifies the perceptive similarity between the original generated image and the adversarial generated image. (3) Multi-Scale Structural Similarity Index Measure (MS-SSIM) (Wang et al., 2003) is a multi-scaled version of SSIM. (4) CLIP (Radford et al., 2021) represents the similarity between the generated image and the text through the CLIP model. (5) Fréchet Inception Distance (FID) (Heusel et al., 2017) quantifies the realism of generated images. (6) Inception Score (IS) (Salimans et al., 2016) also measures the image quality.

Table 2: The transfer attack performance on different target models. The adversarial examples are crafted on SD2-1. The best results are in **bold**. ↓ indicates the lower the better.

| Model | Method | PSNR↓ | SSIM↓ | MS-SSIM↓ | CLIP↓ | FID↑ | IS↓ |
|---|---|---|---|---|---|---|---|
| SD1-5 | Image | 13.38 | 0.300 | 0.499 | 34.32 | 176.42 | 16.41 |
| | Text | 11.88 | 0.232 | 0.388 | **30.06** | 188.65 | **15.83** |
| | MMA | **11.26** | **0.181** | **0.343** | 30.13 | **189.27** | 16.31 |
| SD1-4 | Image | 13.36 | 0.295 | 0.497 | 34.35 | 177.99 | 15.98 |
| | Text | 11.90 | 0.231 | 0.387 | **29.97** | **190.31** | **15.85** |
| | MMA | **11.33** | **0.183** | **0.346** | 30.12 | 189.91 | 15.89 |
| Pix2pix | Image | 14.29 | 0.221 | 0.492 | 34.02 | 154.43 | 16.26 |
| | Text | 13.99 | 0.436 | 0.594 | 32.64 | 123.82 | 16.15 |
| | MMA | **13.16** | **0.216** | **0.460** | **32.45** | **146.57** | **16.06** |

**Parameter:** We conduct the experiments on an A100 GPU server. With the aim of a high imperceptibility, we set the hard constraint of the image perturbation budget $\epsilon = 0.1$ and the maximum token-wise Levenshtein distance of the text to $5$. The iteration number of MMA is $T = 15$. The thresholds for textual perturbation candidates are $\tau_l = 0.3$ and $\tau_h = 0.5$. The step size of the image perturbation is set to be $0.015$. The query time of the image perturbation is set to $5$, and the random mask is sampled from a Bernoulli distribution with a probability of $0.7$. The number of the diffusion step is set to be $15$ for attack and $100$ for inference. We take the strength and guidance of the diffusion models to be $1.0$ and $7.5$ by default.

## 5.2 ATTACKING PERFORMANCE

We first analyze the white-box attacking performance as shown in Table 1. The adversarial sample of MMA is imperceptible because the text and image perturbations are small. MMA only changes less than 3.4 words on average, and the average $L_2$ image distortion is about 10.7. Additionally, around 67% of the image and 84% of the text semantics are preserved. Compared with the single modality version of MMA, MMA can reduce about 27 % of the image perturbation and 13 % of the text perturbation, and better keep the semantics of image and text by around 7% and 4%, respectively. Compared with SOTA single modality attacks, MMA outperforms LDMR on the image perturbation and QF on $lev$. Although the semantic similarity of QF is similar to MMA, the number of modified tokens is 2.86 times more than MMA, which is noticeable. Additionally, the qualitative results of the generated adversarial examples are shown in the first two rows of Figure 3. We can hardly differentiate the difference between images, and the semantics of texts remain the same, demonstrating the low imperceptibility of MMA.

We then analyze the effectiveness of MMA in Table 1. We first compare the MMA with single-modality attacks (Text, Image, and SOTA baselines). MMA outperforms single-modality attacks on PSNR, SSIM, and MS-SSIM under all the experimental settings and is comparable with Text or image modality attacks on other metrics, demonstrating the effectiveness of MMA. Then, we consider the ablation situation (MMA w/o Text and MMA w/o Image). MMA performs better than the ablations of MMA, with a large margin of 0.18 on SSIM and 0.35 on CLIP on average, illustrating the effectiveness of both image and text perturbation. We further compare the single modality attacks with ablations of MMA. Image improves PSNR by 10.5 % more than MMA w/o Text, but it requires 27 % more image perturbations. The finding is similar when it comes to the text modality. Therefore, although the single-modality attack achieves relatively higher performance, the increment in the perturbation outweighs the performance gain. The qualitative results are shown in Figure 3.

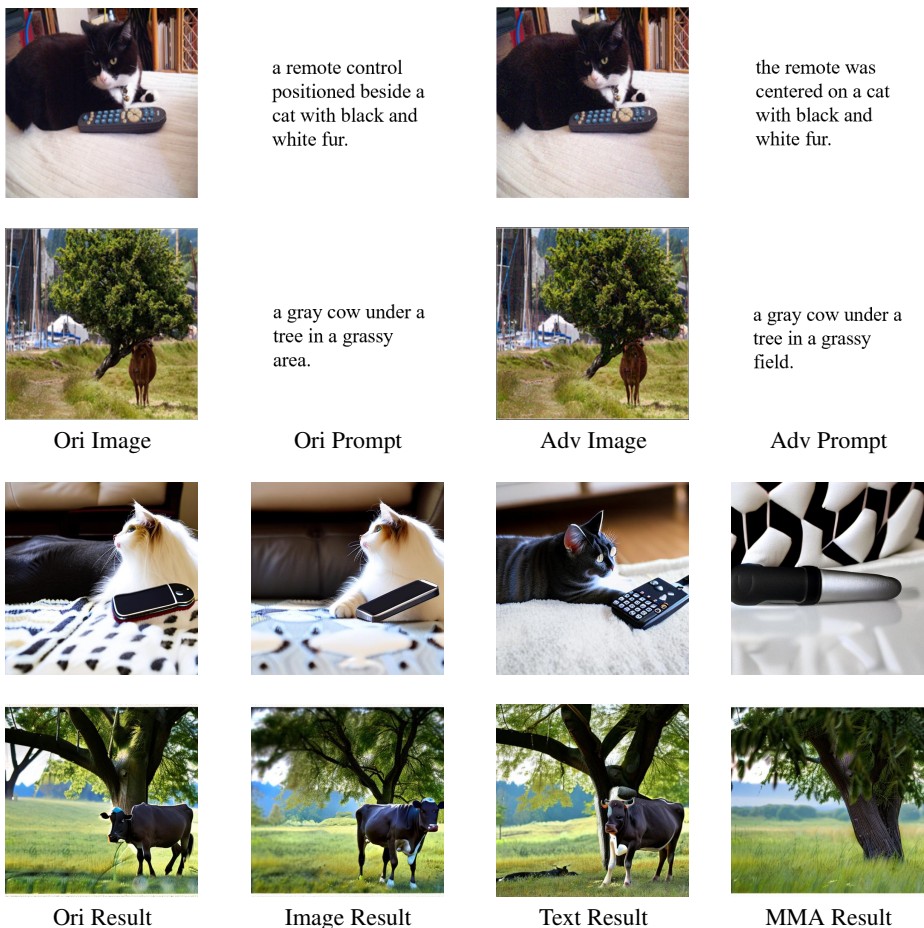

Figure 3: Qualitative results. Row 1 and row 2 show the original and adversarial input pairs. The adversarial inputs are generated by MMA on attacking SD2-1. Row 3 and row 4 illustrate the generated images by different attack methods with the input pairs from row 1 and row 2. All the images are generated by SD2-1.

Perturbing image or text fail to mislead the generation, while MMA can destroy the generation functionality. In conclusion, MMA distributes the perturbation to image and text modality for better attacking performance with better imperceptibility.

After that, we consider the transfer-based black-box setting, where we directly test other LDMs with the adversarial examples generated from a surrogate model. As shown in Table 2, we can draw a similar conclusion that MMA outperforms single-modality attacks on PSNR, SSIM, and MS-SSIM metrics and is comparable with single-modality attacks on other metrics. The experimental results confirm the high transferability of MMA. More quantitative results are shown in the Appendix.

Additionally, we do analysis studies on the MMA to analyze the selection of the attacking modules inside LDM and the distribution of the perturbation from different modalities in the Appendix.

## 6 CONCLUSION

In this paper, we define the multi-modality attacks on LDMs and propose the first multi-modality adversarial attack on LDMs called MMA, which modifies the image and text simultaneously in a unified query ranking framework. The framework includes candidate generation and candidate ranking to combine the text and image perturbation properly. Extensive experiments validate the effectiveness and imperceptibility of MMA.

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

APPENDIX

We supplement the algorithm of candidate generation, more transfer-based black-box experiments, analysis experiments and discuss the efficiency issues of multi-modality attacks on latent diffusion models. Our code is available at `https://anonymous.4open.science/r/Multi-LDM-4087`.

## A  ALGORITHM

The text candidate generation can be summarized in Algorithm 1.

---
**Algorithm 1** Text Candidate Generation (TCG)
---
1: **Input:** input text $t$ and its attribution $A_t$
2: **Input:** the language model $LM$, and two thresholds $\tau_h$ and $\tau_l$
3: **Output:** text candidates $cand_{text}$
4: $cand_{text} = []$
5: **for** $i = 1$ **to** $len(t)$ **do**
6:    $t_{mask} = t.copy()$
7:    $t_{mask}[i] = [MASK]$ ▷ Mask each token
8:    $pred, conf = LM(t_{mask})$ ▷ Token prediction
9:    **if** $A_t[i] > mean(A_t)$ **then**
10:      $threshold = \tau_h$
11:    **else**
12:      $threshold = \tau_l$
13:    **end if**
14:    **for** $j = 1$ **to** $len(conf)$ **do**
15:      **if** $conf[j] > threshold$ **then**
16:        $t_{cand} = t.copy()$
17:        $t_{cand}[i] = pred[j]$
18:        $cand_{text} = cand_{text}.append(t_{cand})$
19:      **end if**
20:    **end for**
21: **end for**
---

The image candidate generation is shown in Algorithm 2.

---
**Algorithm 2** Image Candidate Generation (ICG)
---
1: **Input:** input image $x$ and its attribution $A_x$
2: **Input:** step size $\alpha$
3: **Input:** sampling time $s$, and probability $p$
4: **Output:** image candidates $cand_{img}$
5: $cand_{img} = []$
6: **for** $i = 1$ **to** $s$ **do**
7:    $Mask = Bernoulli(p)$
8:    $img_{cand} = x + \alpha \frac{A_x}{\|A_x\|_2} * Mask$
9:    $cand_{img} = cand_{img}.append(img_{cand})$
10: **end for**
---

## B  TRANSFER PERFORMANCE

As shown in Table 3 and Table 4, we can draw a similar conclusion in the paper that MMA outperforms single-modality attacks on PSNR, SSIM, and MS-SSIM metrics and are comparable with single-modality attacks on other metrics. The experimental result confirms the high transferability of MMA.

Table 3: The transfer attack performance on different target models. The adversarial examples are crafted on SD1-4. The best results are in **bold**. ↓ indicates the lower the better.

| Model | Method | PSNR↓ | SSIM↓ | MS-SSIM↓ | CLIP↓ | FID↑ | IS↓ |
|---|---|---|---|---|---|---|---|
| SD1-5 | Image | 13.07 | 0.296 | 0.483 | 34.20 | 176.74 | 16.23 |
| | Text | 12.00 | 0.247 | 0.404 | **30.70** | 185.46 | 16.58 |
| | MMA | **11.17** | **0.189** | **0.340** | 30.98 | **187.31** | **15.11** |
| SD2-1 | Image | 11.78 | 0.215 | 0.393 | 32.71 | 189.42 | **13.87** |
| | Text | 11.39 | 0.202 | 0.359 | **29.76** | 194.53 | 14.25 |
| | MMA | **10.68** | **0.154** | **0.301** | 29.83 | **198.70** | 13.93 |
| Pix2pix | Image | 14.40 | 0.226 | 0.489 | 34.07 | **156.66** | **15.81** |
| | Text | 14.27 | 0.464 | 0.618 | 32.97 | 124.09 | 17.11 |
| | MMA | **13.51** | **0.220** | **0.465** | **32.65** | 148.40 | 16.16 |

Table 4: The transfer attack performance on different target models. The adversarial examples are crafted on SD1-5. The best results are in **bold**. ↓ indicates the lower the better.

| Model | Method | PSNR↓ | SSIM↓ | MS-SSIM↓ | CLIP↓ | FID↑ | IS↓ |
|---|---|---|---|---|---|---|---|
| SD2-1 | Image | 11.84 | 0.218 | 0.396 | 32.55 | 190.57 | **13.68** |
| | Text | 11.41 | 0.204 | 0.361 | **29.74** | 197.84 | 14.13 |
| | MMA | **10.72** | **0.152** | **0.300** | 29.99 | **200.53** | 14.52 |
| SD1-4 | Image | 13.02 | 0.285 | 0.473 | 33.93 | 176.98 | **15.63** |
| | Text | 12.07 | 0.250 | 0.510 | **30.75** | 186.26 | 15.84 |
| | MMA | **11.29** | **0.190** | **0.348** | 31.15 | **186.75** | 17.02 |
| Pix2pix | Image | 14.34 | 0.223 | 0.484 | 34.03 | **159.13** | 16.22 |
| | Text | **12.75** | 0.264 | 0.532 | 33.10 | 141.35 | 16.62 |
| | MMA | 13.42 | **0.215** | **0.455** | **32.76** | 149.24 | **15.96** |

## C ANALYSIS EXPERIMENT

We first do an ablation study on the attacking module of MMA. In MMA, we consider maximizing the $l_2$ distance of the cross attention module outputs between adversarial input and original input. We consider other modules inside LDMs in the ablation study, including encoder, self-attention, resnet, and decoder. The result is shown in Table 5, and we conclude that attacking the cross attention module is effective under the multi-modality adversarial attack scenario.

Table 5: The ablation study performance on different target modules inside LDM. The adversarial examples are crafted on SD1-5. The best results are in **bold**. ↓ indicates the lower the better.

| Method | PSNR↓ | SSIM↓ | MS-SSIM↓ | CLIP↓ | FID↑ | IS↓ |
|---|---|---|---|---|---|---|
| encoder | 12.87 | 0.289 | 0.471 | 34.18 | 177.67 | **16.29** |
| resnet | 11.91 | 0.268 | 0.419 | 31.67 | 183.98 | 16.67 |
| self-attn | 12.00 | 0.273 | 0.427 | 31.88 | 182.00 | 16.47 |
| decoder | 11.97 | 0.260 | 0.400 | 32.04 | 181.95 | 16.37 |
| cross-attn | **11.17** | **0.185** | **0.337** | **31.16** | **185.96** | 16.47 |

Finally, we analyze the details inside MMA to better understand the multi-modality adversarial attacks. We plot the distribution of the perturbation selection during updating in Figure 4. In the beginning, text perturbation is the dominant choice with more than 60%. Then, the ratio of the image selection increases. The phenomenon also validates that considering both the image and text perturbations is useful.

## D EFFICIENCY ISSUES

We also consider the efficiency issues of MMA. We have to admit that the time complexity of adversarial attacks on diffusion models is intensive because the complexity is related to the denoising

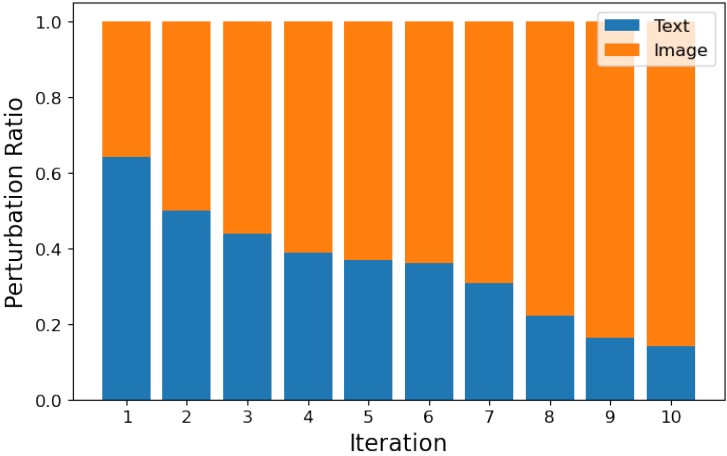

Figure 4: The ratio of the perturbation in each iteration on the whole test set.

step. The general updating should back-propagate through all the denoising steps. In order to cope with the high time complexity, our MMA only updates ten iterations and constrains the search space with proper thresholds. Besides, we only consider 15 denoising steps under the attacking scenario, which can dramatically reduce the time complexity and memory cost. As a result, our MMA is practical, and the time consumption is similar to previous works (Zhang et al., 2023b).

