# OpenReview forum: "Multi-modality Adversarial Attacks on Latent Diffusion Models"
_ICLR.cc/2024/Conference — Submitted to ICLR 2024_

### Official Review · Reviewer_ufds · 2023-10-26

**Soundness:** 2 fair
**Presentation:** 3 good
**Contribution:** 3 good
**Rating:** 6
**Confidence:** 4

**Summary:**

The proposed MMA is a multimodal adversarial attack against img2img latent diffusion models in a white-box and black-box setting. which searches for the most optimal adversarial image and adversarial text iteratively. Compared with the single-modality attacks, MMA can generate adversarial samples with smaller perturbation magnitudes, and mislead the target models successfully.

**Strengths:**

1. The paper is generally well-written and easy to follow.
2. It's the first study of multimodal attack on latent diffusion models, which demonstrates the advantage of multimodal attacks over single modality attacks on img2img tasks.
3. The evaluation metrics in the paper are from two perspectives: imperceptibility and effectiveness, which are better for evaluating the performance of the baselines and MMA.

**Weaknesses:**

1. Related missing references:
[1] Diffusion Models for Imperceptible and Transferable Adversarial Attack
[2] Towards Adversarial Attack on Vision-Language Pre-training Models
2. In the transfer-based black-box setting, there is a lack of comparison with image transfer-based attacks such as TI-FGSM, SINIFGSM, and ... in experiments
3. There is no error analysis. When does the method fail and why? What are the limitations of this method?

**Questions:**

1. what's the difference between the Image and MMA w/o Image, and Text and MMA w/o Text in experiments, please describe in detail.
2. Co-attack is a multimodal adversarial attack designed for vision-language pretraining models, integrating both BERT-attack and PGD methods. Would it be feasible to deploy Co-attack against the LDMs discussed in the paper and subsequently contrast its efficacy with the proposed MMA?
Co-attack: Towards Adversarial Attack on Vision-Language Pre-training Models
3. In the transfer-based attack, can the methods, such as TI-FGSM and DI-FGSM be used to attack the target LDMs?

---

> ### Author Response · Authors · 2023-11-20
>
> Q1: Related work.
>
> We focus on attacking diffusion models. In contrast, [1] uses diffusion models to generate adversarial examples against image classifiers, and [2] focuses on attacking vision-language pre-training models. We have added the discussion in our paper.
>
> Q2: Transfer attacks.
>
> We compare our method with two traditional transfer-based attacks, DIM [3] and SIM [4]. Our approach can outperform the baselines.
>
> White-box on SD2-1
>
> | Methods | $L_2$ | $Sim_{img}$ | PSNR | SSIM | MSSSIM | CLIP |FID | IS |
> | :------: | :-----: | :-----: | :-----: | :-----: | :-----: | :-------: | :----: | :----: |
> | DIM | 32.16 | 0.37 | 10.79 |0.160 |0.302 |31.97 | 196.62 |13.82 |
> | SIM | 33.65 | 0.35 | 10.64 |0.153 |0.294 |31.52 | 197.66 |13.67 |
> | Ours | **16.51** | **0.56** |**10.14** |**0.123** |**0.263** |**28.65** | **205.06** |**13.38** |
>
> Black-box on SD1-5
>
> | Methods | PSNR | SSIM | MSSSIM | CLIP |FID | IS |
> | :------: |  :-----: | :-----: | :-----: | :-------: | :----: | :----: |
> | DIM | 12.82 |0.250 |0.417 |33.86 | 176.77 |**16.70** |
> | SIM | 12.67 |0.233 |0.405 |33.67 | 177.82 |16.24 |
> | Ours | **11.26** |**0.181** |**0.343** |**30.13** | **189.27** |16.31 |
>
>
> Q3: Difference of baselines.
>
> As we introduce in Section 5.1, the four baselines are single-modality attacks in our MMA framework. Suppose we update the adversarial pairs for $T$ iterations by our MMA. We perturb images for $T_1$ iterations and text for $T_2$ iterations. Therefore, $T = T_1 + T_2$. “Image” and “Text” run the MMA framework only on one modality. “Image” only perturbs the input image for $T$ iterations by the MMA framework without changing the input text. Similarly, “Text” only perturbs the input text for $T$ iterations by the MMA framework without changing the input image. “MMA w/o Image” and “MMA w/o Text” mean that we run MMA on both modalities, but we only use the perturbations from one modality. Specifically, “MMA w/o Text” means we only perturb the input image by the generated image perturbation of MMA, which is to perturb the image for $T_1$ iterations. “MMA w/o Image” means we only perturb the input text by the generated text perturbation of MMA, which is to perturb the text for $T_2$ iterations.
>
> Q4: Co-attack.
>
> We compare our method with Co-attack. We can see that our approach achieves better performance.
>
> | Methods | $L_2$ | $Sim_{img}$ | $Lev$ | $Sim_{text}$ | PSNR | SSIM | MSSSIM | CLIP |FID | IS |
> | :------: | :-----: | :-----: | :-----: | :-----: | :-----: | :-----: | :-----: | :-------: | :----: | :----: |
> | Co-Attack | 23.62 | 0.42 | 3.86 | 0.77| 10.44 |0.136 |0.284 |29.32 | 202.27 |13.51 |
> | Ours | **16.51** | **0.56** | **3.78** | **0.80** |**10.14** |**0.123** |**0.263** |**28.65** | **205.06** |**13.38** |
>
> Q5: Transfer setting.
>
> We can incorporate DIM and SIM into our method to improve adversarial transferability. We find that the adversarial transferability is improved.
>
> Black-box on SD1-5
>
> | Methods | PSNR | SSIM | MSSSIM | CLIP |FID | IS |
> | :------: |  :-----: | :-----: | :-----: | :-------: | :----: | :----: |
> | Ours | 11.26 |0.181 |0.343 |30.13 | 189.27 |16.31 |
> | Ours + DIM | 11.18 |0.172 |0.327 |29.97 | 189.76 |16.92 |
> | Ours + SIM | **11.12** |**0.168** |**0.322** |**29.93** | **189.92** |**17.13** |
>
>
> [1] Chen J, Chen H, Chen K, et al. Diffusion Models for Imperceptible and Transferable Adversarial Attack[J]. arXiv preprint arXiv:2305.08192, 2023.
>
> [2] Zhang J, Yi Q, Sang J. Towards adversarial attack on vision-language pre-training models[C]//Proceedings of the 30th ACM International Conference on Multimedia. 2022: 5005-5013.
>
> [3] Xie C, Zhang Z, Zhou Y, et al. Improving transferability of adversarial examples with input diversity[C]//Proceedings of the IEEE/CVF conference on computer vision and pattern recognition. 2019: 2730-2739.
>
> [4] Lin J, Song C, He K, et al. Nesterov accelerated gradient and scale invariance for adversarial attacks[J]. arXiv preprint arXiv:1908.06281, 2019.

---

> ### Author Response · Authors · 2023-11-22
>
> Dear Reviewer,
>
> Thank you for your valuable reviews. Since the discussion phase is going to finish in one day, we look forward to your further feedback about our latest response. Do you still have any concerns about our paper after the rebuttal? Do we address your questions? We are happy to discuss with you in more detail. We greatly appreciate your time and feedback.
>
> Best regards,
>
> Authors

---

### Official Review · Reviewer_gcjf · 2023-10-30

**Soundness:** 3 good
**Presentation:** 2 fair
**Contribution:** 2 fair
**Rating:** 5
**Confidence:** 3

**Summary:**

This paper proposes a multi-modality adversarial attack method called MMA for latent diffusion models, which perturbs both the image and text inputs simultaneously to generate more effective and imperceptible adversarial examples compared to single-modality attacks.

**Strengths:**

The methods are intuitive and easy to implement.

**Weaknesses:**

1. The method in the paper is not particularly innovative and is overly simplistic. The approach, especially the text search mechanism, seems rather basic and lacks novelty. No theoretical analysis is provided on the distortion induced by multi-modality attacks compared to single modality.

2. The paper's primary method appears constrained to untargeted attacks. This limitation curtails its adaptability and potentially its effectiveness in targeted adversarial scenarios.

3. The human evaluation is limited to using CLIP similarity as a proxy. Direct human studies evaluating imperceptibility would be more convincing.

4. The efficiency and computational overhead of the method are not analyzed or compared to baselines.

**Questions:**

See the above weakness.

---

> ### Author Response · Authors · 2023-11-20
>
> Thank you for your valuable review!
>
> Q1: Theoretical analysis.
>
> Compared with single-modality attacks, multi-modality attacks have a lower upper bound of the perturbation. The proof is as follows.
>
> We suppose the perturbation of each step is $(\delta^j_i, \delta^j_t)$ for multi-modality adversarial attacks, where $\delta^j_i$ and $\delta^j_t$ are the perturbation on the image and text respectively.
>
> In our MMA approach, we only update image or text in each step, so $\delta^j_i = \textbf{0}$ or $\delta^j_t = \textbf{0}$.
>
> Without loss of generalization, we focus on the perturbation on the image. Thus, the total perturbation on the image $\Delta_i = \sum_{j}^{T} \delta^j_i$, where $T$ is the total iteration.
>
> Then the $L_2$ norm of the total image perturbation holds the inequality $|| \Delta_i ||_2 = || \delta^1_i + \cdot + \delta^T_i||_2 \leq  || \delta^1_i||_2 + \cdot + || \delta^T_i||_2$. For each perturbation $|| \delta^j_i||_2$, we have the constraint that $|| \delta^j_i||_2 = \alpha$, where is the step length for each iteration.
>
> As a result, the $|| \Delta_i ||_2 \leq \alpha * M$, where $M \leq T$ is the iteration times to update image.
>
> However, for single-modality attack, the total image perturbation $|| \Delta_i ||_2 \leq \alpha * T$.
>
> All in all, multi-modality attacks have a lower upper bound of the perturbation unless the multi-modality attack is degenerated to single-modality attack.
>
> Q2: Targeted attack.
>
> Our approach can be adapted to the targeted adversarial scenario. Specifically, we minimize the distance between the adversarial input pair and the target image-prompt pair in the latent space of the cross-attention module. The results are as follows. Our attack is still effective and imperceptible in targeted adversarial scenarios.
>
> | Methods | $L_2$ | $Sim_{img}$ | $Lev$ | $Sim_{text}$ | PSNR | SSIM | MSSSIM | CLIP |FID | IS |
> | :------: | :-----: | :-----: | :-----: | :-----: | :-----: | :-----: | :-----: | :-------: | :----: | :----: |
> | LDMR| 29.14 | 0.38 | - | - | 9.66 |0.232 |0.425 |32.22 | 292.53 |14.87 |
> | QF| - | - | 9.95 | **0.83**| 10.16 |0.268 |0.484 |32.56 | 189.61 |16.73 |
> | Ours | **16.72** | **0.53** | **3.74** | 0.81 |**8.62** |**0.214** |**0.403** |**31.92** | **203.72** |**14.63** |
>
> Q3: User study.
>
> Please see Q3 of Reviewer FHsv.
>
> Q4: Efficiency.
>
> We compare our time efficiency with baselines. We compute the average time to generate one adversarial example. MMA, LDMR, and QF use 53.2, 49.2, and 44.7 seconds, respectively. Although our time efficiency is similar to the previous works, we can achieve better imperceptibility and attacking performance.

---

> ### Author Response · Authors · 2023-11-22
>
> Dear Reviewer,
>
> Thank you for your valuable reviews. Since the discussion phase is going to finish in one day, we look forward to your further feedback about our latest response. Do you still have any concerns about our paper after the rebuttal? Do we address your questions? We are happy to discuss with you in more detail. We greatly appreciate your time and feedback.
>
> Best regards,
>
> Authors

---

### Official Review · Reviewer_dU51 · 2023-11-01

**Soundness:** 3 good
**Presentation:** 3 good
**Contribution:** 2 fair
**Rating:** 5
**Confidence:** 4

**Summary:**

The paper presents an adversarial attack framework aiming to conduct attacks on the text and image modalities in a unified framework, to reduce the attack magnitude on a single modality and make the attack unnoticeable.

**Strengths:**

The argued first consideration of a unified multi-modal adversarial attack framework.

Good writing and easy to follow.

**Weaknesses:**

lacking deep consideration about the multi-modal generation problem.

The evaluation is not convincing.

**Questions:**

All my concerns come from the main challenge, the core contribution of the paper, all of which are more important and should be clarified rather than the first work on the adversarial attack on multiple modalities, in my opinion. Besides, there also exist some questions.

1. The authors only present a solution for a multi-modal adversarial attack via attacking two modalities and selecting a prominent one. I cannot capture the core problem in the multi-modal adversarial attack and difficulties. Please make a thorough illustration.
2. The evaluation metric designed by the authors is confusing. The paper evaluates PSNR between the original generated image and the adversarial generated image, but I am not sure whether this comparison pair has convincing effects.  I understand the authors aim to estimate the generation effect, but directly comparing two generated images seems to lack stable indications.
3. The adversarial between the two modalities has no relationship in the paper, also verified in Table 1. Under this condition, I think the paper requires further improvement.
4. Please clarify the differences in the utilization of SD-2-1, SD-1-4, and SD-1-5.
5. The imperceptibility has no visible verification.

---

> ### Author Response · Authors · 2023-11-20
>
> Thank you for your valuable review!
>
> Q1: Challenges & contributions.
>
> As we stated in the third paragraph of Section 1, perturbing the text and image at the same time distributes the needed perturbation to both modalities, making them more imperceptible to humans on the whole. Besides, compared with single-modality attacks, multi-modality attacks have the potential to discover more robustness issues. Therefore, the challenge resides on the proper interaction between two modalities to find the optimal perturbation in each step for both good attacking performance and imperceptibility. Therefore, we propose MMA, which uses a unified query ranking framework to properly combine the updating on both modalities.
>
> Furthermore, we are the first to study the multi-modality attacks on diffusion models. We formally define the multi-modality attacks, specifying the attack objective, the perturbation space, and the attack constraints.
>
> Multi-modality attacks also have multiple applications. The generated adversarial examples can be utilized to retrain the model for improving the robustness [3], and to fingerprint the models [4] and watermark the examples [5] for solving privacy issues.
>
> Q2: Evaluation.
>
> We follow previous works [1,2] to utilize PSNR as a metric to quantify the attacking performance. PSNR measures the difference between the generated images of the original input and the adversarial input. In addition to PSNR, we also deploy other metrics, like SSIM, and MSSSIM to measure the difference. The results on these metrics validate the effectiveness of MMA.
>
> Q3: Two modalities.
>
> The two modalities have different influences on the output results. Text perturbations mislead the high-level features and the overall function of image editing, while image perturbations influence the low-level features and reduce the generation quality. As shown in the third row of Figure 3 in the paper, the scenario of the edited image (a cat and a bed) keeps the same under image perturbations, but the text perturbations change the scenario. Besides, as shown in the last row of Figure 3 in the paper, combining perturbations from two modalities can effectively break the normal functionality of diffusion models, while the single-modality perturbation fails to do so.
>
>
> Q4: Model differences.
>
> The difference between SDv1 and SDv2 is on the encoder of the image. SDv1 uses the pre-trained CLIP, while SDv2 retrains CLIP on their own dataset. The difference between SDv1-4 and SD-v1-5 is on the model weights. The latter version is further trained based on the previous version.
>
> Q5: Qualitative examples.
> Please refer to the top 2 rows of Figure 3 in our paper. We can see that both the image and text perturbations are imperceptible to humans.
>
>
> [1] Salman H, Khaddaj A, Leclerc G, et al. Raising the cost of malicious ai-powered image editing[J]. arXiv preprint arXiv:2302.06588, 2023.
>
> [2] Zhang J, Xu Z, Cui S, et al. On the Robustness of Latent Diffusion Models[J]. arXiv preprint arXiv:2306.08257, 2023.
>
> [3] Shafahi A, Najibi M, Ghiasi M A, et al. Adversarial training for free![J]. Advances in Neural Information Processing Systems, 2019, 32.
>
> [4] Peng Z, Li S, Chen G, et al. Fingerprinting deep neural networks globally via universal adversarial perturbations[C]//Proceedings of the IEEE/CVF Conference on Computer Vision and Pattern Recognition. 2022: 13430-13439.
>
> [5] Liang C, Wu X, Hua Y, et al. Adversarial Example Does Good: Preventing Painting Imitation from Diffusion Models via Adversarial Examples[J]. 2023.

---

> ### Author Response · Authors · 2023-11-22
>
> Dear Reviewer,
>
> Thank you for your valuable reviews. Since the discussion phase is going to finish in one day, we look forward to your further feedback about our latest response. Do you still have any concerns about our paper after the rebuttal? Do we address your questions? We are happy to discuss with you in more detail. We greatly appreciate your time and feedback.
>
> Best regards,
>
> Authors

---

### Official Review · Reviewer_FHsv · 2023-11-01

**Soundness:** 2 fair
**Presentation:** 1 poor
**Contribution:** 2 fair
**Rating:** 3
**Confidence:** 4

**Summary:**

This paper introduces the Multi-Modality Adversarial Attack (MMA) algorithm, specifically designed for latent diffusion models (LDMs). MMA simultaneously modifies text and image components within a unified framework, effectively addressing multi-modal robustness. Extensive experiments demonstrate MMA's effectiveness in triggering LDM failures while requiring a smaller perturbation budget compared to single modality attacks, enhancing invisibility.

**Strengths:**

1. Exploration of an intriguing research topic focused on attacking latent diffusion models, shedding light on a relatively unexplored area.
2. Straightforward and easy-to-understand approach to address the challenges associated with the subject matter.

**Weaknesses:**

1. The evaluation metrics for the attack used in the paper cannot be used to quantitatively measure the success rate of the proposed attack. The proposed measurements are all performance metrics related to the image quality of the latent diffusion model and are relative indicators that only show that the attack is working roughly well.
2. The rationale behind the necessity of this attack is not thoroughly convincing. The paper lacks a clear articulation of the contributions that the proposed attack brings to the field of latent diffusion models.
3. The absence of human evaluation metrics to quantitatively assess the effectiveness of the attack in generating adversaries is a notable limitation.
4. The paper does not provide sufficient details regarding the success rate of the attack. Information on how many instances the attack succeeded and the overall success rate is missing, leaving a gap in the evaluation of its performance.
5. The use of citation methods such as \citep and \citet could be improved for better clarity and presentation in the paper, to distinguish between citations with or without parentheses.

**Questions:**

NA

---

> ### Author Response · Authors · 2023-11-20
>
> Thank you for your valuable review!
>
> Q1: Measurement.
>
> A diffusion model is a generative model. Therefore, attacking diffusion models is different from the classic adversarial attacking scenario, which targets image classifiers and has an explicit attacking objective and measurement as we discussed in Section 3. Therefore, we cannot quantify the performance of the attack by the success rate. Instead, we utilize the performance metrics in the generative model literature [1,2], which allows us to quantify the attacking performance by the generation quality and the similarity between the generated images and prompts under attacks. Besides, the selected metrics are widely used in previous studies of adversarial attacks on diffusion models [3,4].
>
>
> Q2: Contribution.
>
> Adversarial attacks can be used to evaluate the robustness of the target model. As we discussed in the third paragraph of Section 1, we need multi-modality attacks for the below reasons.
>
> (1)	Compared with the single-modality attacks on diffusion models, multi-modality attacks can more thoroughly explore the perturbation space. Therefore, multi-modality attacks can find more failures of diffusion models than single-modality attacks.
>
> (2)	Perturbing the text and image at the same time distributes the needed perturbation to both modalities, making them more imperceptible to humans on the whole.
>
> (3)	The generated adversarial examples can also be utilized to retrain the model for improving the robustness [5], and to fingerprint the models [6] and watermark the examples [7] for solving privacy issues.
>
> Q3: User study.
>
> In order to measure the success of the attack and its imperceptibility, we launch a preliminary user study, where we randomly select 100 examples, and ask three annotators to give their scores. We compare our approach with baselines LDMR and QF. For measuring the ASR, we ask the annotator the following question: How well does the generated image correspond with the prompt? Please score it from 1 to 5, where the higher score means a higher consistency. For measuring the imperceptibility, we ask the annotator the following question: How similar are the two data pairs (the adversarial pair and the original pair)? Please score it from 1 to 5, where the higher score means a higher similarity.
>
> The results are shown in the table below. The agreement of the user study is 80.7 %. We can see that our approach achieves the best attacking performance and imperceptibility.
>
> | Methods |Consistency (ASR) | Similarity (Imperceptibility) |
> | :------: | :----: | :----: |
> | LDMR| 3.52 | 2.87 |
> | QF| 3.14 | 2.15 |
> | Ours | **2.47** | **4.58** |
>
>
>
> Q4: ASR.
>
> As we mentioned in Q1, the attack success rate is not suitable for quantifying the performance of attacks on diffusion models. If needed, we can set a threshold on the drop of the CLIP score to represent whether the attack is successful or not. We set the threshold to be 2.0. The ASR of MMA is 99.6\%, while the ASR of LDMR and QF are 10.5\% and 21.3\%, respectively. Our approach outperforms the baselines with a large margin on ASR.
>
> Q5: Presentation.
>
> The citation in the paper is modified as you suggested.
>
> [1] Ho J, Jain A, Abbeel P. Denoising diffusion probabilistic models[J]. Advances in neural information processing systems, 2020, 33: 6840-6851.
>
> [2] Rombach R, Blattmann A, Lorenz D, et al. High-resolution image synthesis with latent diffusion models[C]//Proceedings of the IEEE/CVF conference on computer vision and pattern recognition. 2022: 10684-10695.
>
> [3] Salman H, Khaddaj A, Leclerc G, et al. Raising the cost of malicious ai-powered image editing[J]. arXiv preprint arXiv:2302.06588, 2023.
>
> [4] Zhang J, Xu Z, Cui S, et al. On the Robustness of Latent Diffusion Models[J]. arXiv preprint arXiv:2306.08257, 2023.
>
> [5] Shafahi A, Najibi M, Ghiasi M A, et al. Adversarial training for free![J]. Advances in Neural Information Processing Systems, 2019, 32.
>
> [6] Peng Z, Li S, Chen G, et al. Fingerprinting deep neural networks globally via universal adversarial perturbations[C]//Proceedings of the IEEE/CVF Conference on Computer Vision and Pattern Recognition. 2022: 13430-13439.
>
> [7] Liang C, Wu X, Hua Y, et al. Adversarial Example Does Good: Preventing Painting Imitation from Diffusion Models via Adversarial Examples[J]. 2023.

---

> ### Author Response · Authors · 2023-11-22
>
> Dear Reviewer,
>
> Thank you for your valuable reviews. Since the discussion phase is going to finish in one day, we look forward to your further feedback about our latest response. Do you still have any concerns about our paper after the rebuttal? Do we address your questions? We are happy to discuss with you in more detail. We greatly appreciate your time and feedback.
>
> Best regards,
>
> Authors

---

### Meta-Review · Area_Chair_Hff6 · 2023-12-07

**Metareview:**

The reviewers raised multiple concerns, including the novelty of the multimodal attack strategy, baseline methods, imperceptibility visualization and evaluation metrics. After the rebuttal, the reviewers do not give proper feedback in time. I myself have taken a closer look at the paper as well as the review threads. I find that although the manuscript has been largely improved through the author feedback, the main contribution of this paper is not clearly presented in this version, as pointed out by Reviewer FHsv, dU51 and gcjf.

More details of the core contribution from my side: Since the single modality attacks were already proposed before, the main tech contribution of the multi-modal attack on LDM seems to lie in the ranking part (sec 4.2). This ranking should be explained in detail and compared with the previous multi-modal attacking methods, such as co-attack, either compared via comments or empirical results.

**Justification For Why Not Higher Score:**

The main contribution of this paper is not clearly presented in this version, as pointed out by Reviewer FHsv, dU51 and gcjf.

**Justification For Why Not Lower Score:**

n/a

---

### Decision · Program_Chairs · 2024-01-16

Reject